

# Antimicrobial resistance and molecular typing of *Staphylococcus aureus* isolates from raw milk in Hunan Province

Keming Ning[1],Rushun Zhou[2] and Manxiang Li[1]

[1] College of Veterinary Medicine, Hunan Agricultural University, Hunan, China
[2] Hunan Provincial Institute of Veterinary Drugs and Feed Control, Hunan, China

## ABSTRACT

**Background.** *Staphylococcus aureus* is one of the most important foodborne pathogens in the world and the main cause of dairy cow mastitis. Few studies have investigated the epidemic pedigree of *S. aureus* of bovine origin in Hunan, China. Therefore, we aimed to analyze the capsular polysaccharides (CP), molecular typing, and antibiotic resistance characteristics of *S. aureus* isolated from raw milk of dairy farms in Hunan Province.

**Methods.** Between 2018 and 2022, 681 raw milk samples were collected from dairy cows from farms in Changsha, Changde, Shaoyang, Yongzhou, and Chenzhou in Hunan Province. *S. aureus* was isolated from these samples, and the isolates were subjected to molecular typing, CP typing, and determination of antibiotic resistance through broth dilution and polymerase chain reaction (PCR).

**Results.** From 681 raw milk samples, 76 strains of *S. aureus* were isolated. The pathogenicity of 76 isolates was determined preliminarily by detecting *cp5* and *cp8* CP genes. Eighteen types of antimicrobial resistance phenotypes of 76 *S. aureus* strains were detected by the broth dilution method, and 11 kinds of related resistance genes were amplified by PCR. The *S. aureus* isolates had CP5 (42.10%) and CP8 (57.89%). *S. aureus* had a multiple antimicrobial resistance rate of 26.75%. The isolated strains had the highest resistance rate to penicillin (82.89%) and showed varying degrees of resistance to other drugs, but no isolate showed resistance to doxycycline. The 76 isolates all carried two or more antibiotic resistance genes, with a maximum of eight antibiotics resistance genes. *FemB* was detected in all isolates, but none of isolates carried *vanA*, *ermA*, or *glrA*. The 76 isolates were divided into 22 sequence types (ST) and 20 *spa* types by MLST and *spa* typing, and the number of t796-ST7 ($n = 15$) isolates was the highest, which may be the major epidemic strain of multidrug-resistant *S. aureus*.

**Conclusion.** The present findings indicate the need to increase production of the CP8 *S. aureus* vaccine in Hunan Province and strengthen resistance monitoring of t796-ST7 isolates with the prevalent molecular type of multi-drug resistant strains. The use of $\beta$-lactam, macrolides, and lincosamides should be reduced; doxycycline, sulfonamides, and glycopeptides could be appropriately added to veterinary antibiotics to treat infectious diseases in dairy cows.

Corresponding author
Manxiang Li, manxiangl@163.com

## INTRODUCTION

*Staphylococcus aureus* has long been a threat to human health and the development of animal husbandry. In many countries, 30% to 40% cases of bovine mastitis are caused by *S. aureus* (*Basanisi et al., 2017*), mainly through adhesion, which leads to colonization in the mammary epithelial cells of livestock and results in udder abscesses, hardening, fistulae, necrosis, and exfoliation. Diseased bovines present with obvious udder fever, udder hardness and pain, and reddish to black milk with foul-smelling flocs, which not only reduces milk production and quality, but also endangers the animal's life (*Hiramatsu et al., 2002*; *Sinha & Fraunholz, 2010*). Non-standard processing and disinfection procedures can easily result in the production of dairy products contaminated with *S. aureus* (*Schmidt, Kock & Ehlers, 2017*; *Johler et al., 2018*), and many types of *S. aureus* can be detected in raw dairy products worldwide (*Alghizzi & Shami, 2021*). *S. aureus* causes infection in the host body by releasing a variety of toxins, the most important being Panton–Valentine leukocidin (PVL). Encoded by *lukF PV* and *lukS PV*, PVL is a poor-forming toxin that causes host immune suppression and infection by attacking leukocytes, such as neutrophils and monocytes (*Spaan, Van Strijp & Torres, 2017*); *S. aureus* exfoliative toxin (Ets) causes severe inflammation of the gastrointestinal tract by stimulating intestinal epithelial cells (*Banwell & Sherr, 1973*). Toxic shock syndrome toxin can promote activation of T lymphocytes and enhance the secretion of inflammatory factors, resulting in a severe immune response of the animal body with subsequent multiple organ failure (*Hoppe et al., 2018*). Ets reduces the adhesion of mammalian stratum corneum cells by acting on desmosomal core glycoproteins between surface granular cells, resulting in the loosening of keratinocytes, which leads to the staphylococcal scalded-skin syndrome. Bacteria enter the bloodstream, leading to septicemia and endocarditis as the immunity of the infected host decreases (*Rodríguez-Lázaro et al., 2015*). Recently, foodborne *S. aureus* infections have been frequently reported in China (*Wu et al., 2019*). In 2015, microbial food poisoning accounted for 53.7% of all food poisoning emergencies in China, and *S. aureus* was the main pathogen (*Wu et al., 2018*). *S. aureus* not only has a great impact on the dairy industry but also poses a threat to public health and food safety.

The invasiveness of *S. aureus* depends on its resistance to host phagocytes and its cell adhesion. The capsular polysaccharide (CP) of *S. aureus* is an important factor related to its anti-phagocytosis and cell adhesion properties, and it can be classified into 11 serotypes based on its antigenic structure (*Sompolinsky et al., 1985*). CP1 and CP2 are rarely encountered in clinical practice, and approximately 70% to 80% of *S. aureus* can produce CPs of serotypes 5 and 8, referred to as CP5 and CP8, respectively (*Goerke et al., 2005*). CP5 and CP8 are expressed by the *cp5* and *cp8* genes, respectively. *Cp5* and *cp8* are nucleic acid sequences of approximately 17.5 kb present on bacterial chromosomes. The structures of CP5 and CP8 are 14-3-O-Ac- $\beta$-D-Man-NAcA-(1-4)-lFucNAc-(1-3)- $\beta$-D-FucNAc-(1-)n and(3)-4-O-Ac- $\beta$-D-ManNAcA-(1-3)-L-FucNAc-(1-3)- $\beta$-D-FucNAc-(1-), respectively. The structures of CP5 and CP8 mainly differ in terms of the position of sugar linkage and acetylation of amino mannituronic acid residues. It has been shown that CP5 and CP8 are the main factors associated with *S. aureus*-induced mastitis (*Poutrel et al., 1988*), and

serotyping of *S. aureus* CPs is of significance to the local dairy industry to better understand the epidemiology of *S. aureus*.

At present, the main method for preventing and treating dairy cow mastitis caused by *S. aureus* is udder infusion with antibiotics. However, with the extensive use of antibiotics, *S. aureus* has acquired resistance to a variety of antibiotics by changing its phenotype, regulating metabolic pathways, and horizontally transferring resistant plasmids. Methicillin-resistant *S. aureus* (MRSA) has spread across the globe since its discovery. Despite its resistance to new semisynthetic *β*-lactams, MRSA is resistant to a variety of antibiotics, such as macrolides, aminoglycosides, and tetracyclines (*Hassoun, Linden & Friedman, 2017*). In the 1990s, MRSA infection, acquired immune deficiency syndrome, and hepatitis B were considered the three most difficult diseases to treat worldwide (*Nada et al., 1996*). In the animal husbandry industry, MRSA was first isolated from dairy cows with mastitis in 1972 (*Devriese, Van Damme & Fameree, 1972*), and cows have become its natural hosts (*Spohr et al., 2011*). The penicillin-binding protein (PBP) 2a encoded by *mecA* is the main cause for MRSA, and related studies have indicated that the detection of *mecA* is the main criterion for identifying MRSA (*Kobayashi et al., 1994*). Oxacillin-sensitive *mecA*-positive *S. aureus* (OS-MRSA) has been detected in human clinical and animal products (*Conceição et al., 2015*). Although OS-MRSA carries *mecA*, it is often misidentified as methicillin-sensitive *S. aureus* (MSSA) (*Quijada et al., 2019*) and is likely to become a subgroup with high resistance under the pressure of *β*-lactam selection (*Martineau et al., 2000*). Vancomycin is considered the last line of defense against Gram-positive bacteria and is used to treat infections that do not respond to other antibiotics. However, vancomycin-resistant *S. aureus* (VRSA) has become increasingly common in China and abroad (*Haseeb et al., 2019*). The resistance of *S. aureus* to vancomycin is usually mediated by the *vanA* operon on transposable Tn1546, which results from the transfer of resistant plasmids from vancomycin-resistant enterococci (*Arthur et al., 1993*; *Lazaris et al., 2017*). Under the combined action of *vanA* and *vanH* expression products, d-Lac replaces d-Ala in the cell wall peptidoglycan precursor peptide to form d-Ala-d-Lac, which alters the vancomycin binding site and leads to drug failure (*Bugg et al., 1991*). As the incidence of multidrug resistance has increased all over the world and is considered a significant public health threat, the proper use of antibiotics has become even more necessary. Moreover, the emergence of multidrug-resistant (MDR) and extensively drug-resistant(XDR) strains greatly increases the risk of the resistance spreading to humans and having an impact on the effectiveness of current antibiotic treatments; in fact, this is often a major factor in the failure of treatment and infection control (*Algammal et al., 2020*; *Helmy et al., 2018*). Additionally, studies have shown that MDR MRSA isolated from milk carries a variety of major virulence factors, such as *sea* and *coa*, that can lead to food poisoning in consumers who ingest the contaminated milk (*Algammal et al., 2020*). The misuse of antibiotics is an important factor leading to the emergence of MDR and XDR pathogens that threaten human public health as well as livestock-derived and aquatic products (*Algammal et al., 2022*). Therefore, effective and rigorous monitoring methods need to be established to control their growth in humans and animals (*Algammal et al., 2023*) and for routine application of antimicrobial susceptibility testing to detect the antibiotic of choice and

screen emerging resistant strains. In this context, monitoring the antimicrobial resistance of *S. aureus* and prevalence of MRSA and VRSA in raw milk is of great significance for the research and development of antibiotics and for identifying the most appropriate drug treatment for animal diseases from the perspective of food safety.

Multilocus sequence typing (MLST) is based on the technology of multilocus enzyme electrophoresis. It involves seven pairs of relatively conservative housekeeping genes in the bacterial genome that are sequenced directly and uploaded to an international public database for analysis, which greatly improves the accuracy of multilocus enzyme electrophoresis (*Maiden et al., 1998*). In addition to the advantages of general genotyping methods, MLST has high resolution and repeatability, making it more convenient and accurate than general genotyping methods (*Tivendale, Allen & Browning, 2009*). More importantly, MLST overcomes the space restrictions of laboratories, regions, and countries, helps share typing data through large international databases, and enables analysis of the phylogeny, population structure, biological evolution, epidemiology, and molecular genetic evolution of pathogenic microorganisms (*Van Wamel et al., 2010*). *spa* typing is used to sequence the X variable region gene of *S. aureus* protein A; it also reveals changes in base pair numbers, sequence of X region repeat succession, and actions, such as base insertion and deletion. The X region is polymorphic; therefore, the isolates are typed according to the polymorphism of this region (*Moosavian, Baratian Dehkordi & Hashemzadeh, 2020*). This typing method based on single-base sequence analysis has a standardized international nomenclature and can be integrated into the international *spa* database, which is convenient for exploring molecular correlations between *S. aureus* from different regions, environments, and sources (*Rodriguez et al., 2015*). *spa* typing can usually be mapped to MLST in the form of *spa*-MLST-mapping (*Strommenger et al., 2008*), which has become the main method for typing *S. aureus*. It makes the research results highly reproducible and the results of regional and international *S. aureus* pedigree monitoring and source tracing interchangeable (*Gurjar et al., 2012*).

Consumption of contaminated milk is an important way by which *S. aureus* can infect humans, and several cases of bacterial food poisoning related to *S. aureus* have been reported (*Fetsch et al., 2014*). The fact that raw milk is particularly vulnerable to *S. aureus* contamination, coupled with the increasingly serious antimicrobial resistance of *S. aureus*, makes it a potential threat to the development of the dairy industry and public health safety. Milk forms a significant component of the Chinese balanced diet standards, and the consumption of milk is increasing annually (*Qian et al., 2011*). As the dairy industry in Hunan Province has advanced, large dairy farms have been built in many cities and counties. However, few studies have examined *S. aureus* isolates from raw milk in Hunan. The purpose of this study was to determine the antibiotic resistance, serotype distribution, and molecular epidemic characteristics of *S. aureus* isolated from raw milk of dairy cows in Hunan Province and provide a theoretical basis for rational use of antibiotics in dairy cows in Hunan. These findings could help promote animal welfare and food safety management through rational drug use in the dairy industry.

## MATERIALS & METHODS

### Sample collection

Between 2018 and 2022, 681 raw milk samples were collected from dairy cows from farms in Changsha, Changde, Shaoyang, Yongzhou, and Chenzhou in Hunan Province. The sample collection was approved by the Ethics Committee of Hunan Agricultural University (no. 2022092389). Dairy cow teats were disinfected with cotton balls containing 75% alcohol before milk collection, and the first three handfuls of milk were discarded. All samples were transferred to sterile tubes and transported to the laboratory for bacterial analysis at 4 °C.

### Isolation and identification of *S. aureus*

One milliliter of the collected raw milk sample was added to nine mL of 10% sodium chloride trypticase soy broth (Luqiao Bio-Technology Co., Ltd., Beijing, China) and incubated at 36 °C for 24 h. An aseptic inoculation ring was dipped in the culture, and chromogenic *S. aureus* agar plates (Hope Bio-Technology Co., Ltd, Qingdao, China) were inoculated and incubated at 36 °C for 18 h. Colonies suspected to be of *S. aureus* based on their morphology were selected and subcultured in nutrient agar medium (Huankai Bio-Technology Co., Ltd, Guangdong, China) at 36 °C for 24 h and subsequently incubated at 36 °C for 18 h. Individual colonies on nutrient agar were selected, and catalase tests, Gram staining, matrix-assisted laser desorption ionization-time of flight mass spectrometry (Bruker Daltonics, Karlsruhe, Germany), and VITEK-2 Compact testing(bioMérieux, Marcy-l'Étoile, Lyon, France) were performed. All strains were stored in 20% glycerin broth at −80 °C. The antibiotic-sensitive quality control strain *S. aureus* ATCC 29213 was kindly provided by the Hunan Provincial Institute of Veterinary Drugs and Feed Control.

### Molecular identification of the isolates

Genomic deoxyribonucleic acid (DNA) was extracted using a bacterial DNA extraction kit (Bioer Bio-Technology Co. Ltd., Hangzhou, China) according to the manufacturer's instructions. The isolates were confirmed by polymerase chain reaction (PCR) using a primer pair of the thermonuclease gene (*nuc*) (Table 1) and 2 × Phanta Max Master Mix (Dye Plus)(Vazyme Biotech Co., Ltd, Nanjing, China). The amplified PCR products were separated by 1.0% agarose gel electrophoresis. Then, the PCR products of five strains were randomly selected and sequenced, and the sequencing results were compared with those on the National Center for Biotechnology Information database (http://blast.ncbi.nlm.nih.gov).

### Typing of *S. aureus* CPs

CP5 (*cp5*) and CP8 (*cp8*) genes were amplified using PCR. The primer sequences, amplicon sizes, and annealing temperatures of *cp5* and *cp8* are shown in Table 1.

### Determination of phenotypic antimicrobial resistance

The antimicrobial resistance phenotypes ($n = 18$) of the *S. aureus* isolates were detected using the broth dilution method. Resistance to the following antimicrobial agents was evaluated: $\beta$-lactam antibiotics (penicillin, benzoxicillin, cefoxitin, ceftiofurme, and ogmentine); macrolides (erythromycin and timicacin); lincoamines

**Table 1** Sequences of primer sets with their corresponding polymerase chain reaction protocols and product sizes.

| Target gene | Primer sequence (5′—3′) | Amplicon size (bp) | Annealing temperature (°C) |
|---|---|---|---|
| *nuc* | GCGATTGATGGTGATACGGTTAAAT | 294 | 56 |
|  | AACATAAGCAACTTTAGCCAAGCC |  |  |
| *cp5* | GTCAAAGATTATGTGATGCTACTGAG | 361 | 62 |
|  | ACTTCGAATATAAACTTGAATCAATGTTATACAG |  |  |
| *cp8* | GCCTTATGTTAGGTGATAAACC | 173 | 62 |
|  | GGAAAAACACTATCATAGCAGG |  |  |
| *mecA* | TGGCTCAGGTACTGCTATCC | 533 | 56 |
|  | CACCTTGTCCGTAACCTGAA |  |  |
| *femA* | TGCTGGTAATGATTGGTT | 498 | 56 |
|  | ATCTCGCTTGTTATGTGC |  |  |
| *femB* | TTACAGAGTTAACTGTTACC | 651 | 55 |
|  | ATACAAATCCAGCACGCTCT |  |  |
| *ermA* | TCTAAAAAGCATGTAAAAGAA | 645 | 45 |
|  | CTTCGATAGTTTATTAATATTAG |  |  |
| *ermB* | GAAAAGGTACTCAACCAAATA | 639 | 52 |
|  | AGTAACGGTACTTAAATTGTTTAC |  |  |
| *ermC* | GCTAATATTGTTTAAATCGTCAATTCC | 572 | 50 |
|  | GGATCAGGAAAAGGACATTTTAC |  |  |
| *tetM* | AGTGGAGCGATTACAGAA | 158 | 55 |
|  | CATATGTCCTGGCGTGTCTA |  |  |
| *aacA-aphD* | CCAAGAGCAATAAGGGCATA | 220 | 56 |
|  | CACTATCATAACCACTACCG |  |  |
| *glrA* | AGGGAAGTGTTTCAGTCT | 457 | 55 |
|  | CCATTCTCAATAATACCG |  |  |
| *VanA* | GGGAAAACGACAATTGC | 885 | 55 |
|  | GTACAATGCGGCCGTTA |  |  |
| *Flor* | TGCCAGCAGTGCCGTTTAT | 900 | 55 |
|  | CACCGCCCAAGCAGAAGTA |  |  |
| *arcC* | TTGATTCACCAGCGCGTATTGTC | 456 | 55 |
|  | AGGTATCTGCTTCAATCAGCG |  |  |
| *aroE* | ATCGGAAATCCTATTTCACATTC | 456 | 55 |
|  | GGTGTTGTATTAATAACGATATC |  |  |
| *glpF* | CTAGGAACTGCAATCTTAATCC | 465 | 55 |
|  | TGGTAAAATCGCATGTCCAATTC |  |  |
| *gmk* | ATCGTTTTATCGGGACCATC | 429 | 55 |
|  | TCATTAACTACAACGTAATCGTA |  |  |
| *pta* | GTTAAAATCGTATTACCTGAAGG | 474 | 55 |
|  | GACCCTTTTGTTGAAAAGCTTAA |  |  |
| *tpi* | TCGTTCATTCTGAACGTCGTGAA | 402 | 55 |

**Table 1** (*continued*)

| Target gene | Primer sequence (5′—3′) | Amplicon size (bp) | Annealing temperature (°C) |
|---|---|---|---|
| *yqiL* | TTTGCACCTTCTAACAATTGTAC CAGCATACAGGACACCTATTGGC CGTTGAGGAATCGATACTGGAAC | 516 | 55 |
| *spa* | TAAAGACGATCCTTCGGTGAGC CAGCAGTAGTGCCGTTTGCTT | Variation | 58 |

(clindamycin); quinolones (enrofloxacin and ofloxacin); tetracycline (doxycycline); aminoglycosides (gentamicin); sulfonamides (sulfaisoxazole and compound xazole); glycopeptides(vancomycin); and those of other classes (taymycin and linezolid). Circular or diffuse deposits were observed in the wells with bacterial growth, while no turbidity was observed in the liquid in the negative control wells. Antimicrobial resistance was evaluated based on the minimum inhibitory concentration of an antibacterial drug according to the standard values published by the Clinical and Laboratory Standards Institute in 2022 (https://clsi.org/). *S. aureus* ATCC 29213 was used for quality control in the drug sensitivity test. As described by *Magiorakos et al. (2012)*, the isolates were classified as MDR (resistant to three of the aforementioned categories of drugs), XDR (only sensitive to class 1 and 2 antibiotics) and pandrug-resistant (PDR, resistant to all types of antibiotics).

## PCR amplification of antimicrobial resistance genes

Antimicrobial resistance genes were detected using purified *S. aureus* DNA. The genes for $\beta$-lactam (*mecA*, *femA*, *femB*), macrolide (*ermA*, *ermB*, *ermC*), tetracycline (*tetM*), aminoglycoside (*aacA-aphD*), quinolone (*glrA*), phenicol (*flor*), and vancomycin (*vanA*) resistance were amplified using PCR. The resistance gene primers and annealing temperatures are listed in Table 1.

## MLST and cluster analysis

According to the distribution of alleles and sequence types (STs) in the MLST database (https://pubmlst.org/organisms/staphylococcus-aureus/primers), seven housekeeping genes of *S. aureus* were selected for PCR amplification: carbamate kinase (*arcC*), shikimate dehydrogenase (*aroE*), glycerol kinase (*glpF*), guanylate kinase (*gmk*), phosphate acetyltransferase (*pta*), triosephosphate isomerase(*tpi*), and acetyl coenzyme A acetyltransferase (*yqiL*). The sizes of each gene fragment varied from 400 to 520 bp. The gene primers, amplicon sizes, and annealing temperatures are shown in Table 1. All PCR products were sequenced, and the sequencing results were submitted to the MLST database (https://pubMLST.org/) to obtain the corresponding allelic profiles and STs. Cluster analysis was conducted using the PHYLOViZ online tool (https://online.phyloviz.net/index) and BioNumerics software (version 7.6; Applied-Maths, Sint-Martens-Latem, Belgium).

## *spa* typing

*spa* types of all isolates were assayed using PCR according to the *spa* primers provided in the Ridom *Spa* Server database (https://spa.ridom.de/) (Table 1). All PCR products were

sequenced, the sequence number of the repeat succession was analyzed using Ridom *Spa* Server, and the *spa* types of the isolates were obtained.

### Simpson's index of diversity calculation

The MLST and *spa* types were inputted into the Comparing Partitions online tool (http://www.comparingpartitions.info/index.php?link=Tool) to calculate the Simpson's index of diversity (*Sun et al., 2019*).

### Statistical analyses

The K independent samples method and a non-parametric test were used to identify the correlation between MLST and *spa* typing, antibiotic resistance genes and phenotypes, and to differentiate between the MRSA and molecular types. Statistical analyses were performed using IBM SPSS Statistics for Windows, version 23.0 (IBM Corp., Armonk, NY, USA). ($p < 0.05$ was considered to indicate statistical significance; $p > 0.1$ was not statistically significant)

## RESULTS

Determining the main prevalent molecular types of *S. aureus* and their antibiotic resistance is necessary to formulate effective antibiotic treatment strategies and reduce the impact of *S. aureus* infections on the dairy industry and public health. We therefore examined the CPs, molecular types, and antibiotic resistance characteristics of *S. aureus* isolated from raw milk of dairy farms in Hunan Province.

### Isolation and identification of *S. aureus* from raw milk

Seventy-six strains (11.16%) of suspected *S. aureus* were isolated from 681 raw milk samples collected from dairy herds in five regions of Hunan Province from 2018 to 2022. The isolation rates of *S. aureus* for samples from Changsha, Changde, Shaoyang, Chenzhou, and Yongzhou were 15.78% (12/76), 44.73% (34/76), 32.89% (25/76), 3.94% (3/76), and 2.63% (2/76), respectively. All isolates appeared as smooth, raised, purplish-red, round colonies on chromogenic medium and smooth, golden-yellow, round colonies on nutrient agar, ranging in size from a pinhead to grain of rice, with positive catalase test results. Gram-positive strains arranged in the shape of grapes were observed on microscopic examination. *Nuc* could be detected in all isolates, and the sequence homology between the PCR products and published strains was 99.21% to 99.23%. Matrix-assisted laser desorption ionization-time of flight mass spectrometry and VITEK-2 Compact testing revealed that the isolates were of *S. aureus*. The biochemical reactions and their results are shown in Table 2. Based on the above results, the 76 isolates were consistent with the biochemical characteristics of *S. aureus* and were identified as *S. aureus*.

### CP types of *S. aureus*

All isolates were classified as having CP5 or CP8. The detection rates of CP5 for samples from Changsha, Changde, Shaoyang, Yongzhou, and Chenzhou were 75% (9/12), 35.29% (12/34), 40% (10/25), 0.00% (0/2), and 33.33% (1/3), respectively, while those of CP8 were 25% (3/12), 64.70% (22/34), 60% (15/25), 100% (2/2), and 66.66% (2/3), respectively (Fig. 1).

**Table 2  Results of VITEK 2 automatic microbial biochemical identification.**

| Details of biochemical reaction | | | | | | | | | | | |
|---|---|---|---|---|---|---|---|---|---|---|---|
| AMY | − | PIPLC | − | dXYl | − | ADH1 | + | BGAL | − | AGLU | + |
| APPA | − | CDEX | − | AspA | − | BGAR | − | AMAN | − | PHOS | + |
| LeuA | − | ProA | − | BGURr | − | AGAL | − | PyrA | + | BGUR | − |
| AlaA | − | TyrA | − | dSOR | − | URE | − | POLYB | + | dGAL | + |
| dRIB | − | ILATk | + | LAC | − | NAG | − | dMAL | + | BACI | + |
| NOVO | − | NC6.5 | + | dMAN | + | dMNE | − | MBdG | − | PUL | − |
| dRAF | − | O129R | + | SAL | − | SAC | − | dTRE | + | ADH2s | − |
| OPTO | + | | | | | | | | | | |

**Notes.**

AMY, Amygdalin; PIPLC, Phosphatidyl phospholipase C; dXYl, D-xylose; ADH1, Arginine dihydrolase 1; BGAL, $\beta$-D-galactosidase; AGLU, $\alpha$-glucosidase; APPA, Alanine-phenylalanine-proline arylaminase; CDEX, Cyclodextrin; AspA, L-aspartate arylaminase; BGAR, $\beta$-galactose pyranosidase; AMAN, $\alpha$-mannosidase; PHOS, Phosphatase; LeuA, Leucine arylaminase; ProA, L-proline arylaminase; BGURr, $\beta$-glucosidase; AGAL, $\alpha$-galactosidase; PyrA, Pyroglutamate arylaminase; BGUR, $\beta$-D-glucosidase; AlaA, Alanine arylaminase; TyrA, Tyrosine arylaminase; dSOR, D-sorbitol; URE, Urease; POLYB, Polymyxin B tolerance; dGAL, D-galactose; dRIB, D-ribose; ILATk, L-Lactate produces alkali; LAC, Lactose; NAG, N-acetyl-D-glucosamine; dMAL, D-maltose; BACI, Bacitracin tolerance; NOVO, Neomycin tolerance; NC6.5, 6.5% sodium chloride growth; dMAN, D-mannitol; dMNE, D-Mannose; MBDG, Methyl-B-D-glucopyranoside; PUL, Pullulan; dRAF, D-raffinose; O129R, O/129 tolerance; SAL, Salicin; SAC, Sucrose; dTRE, D-trehalose; ADH2s, Arginine dihydrolase 2; OPTO, Optokhin tolerance.

+ Reaction positive; −Reaction negative

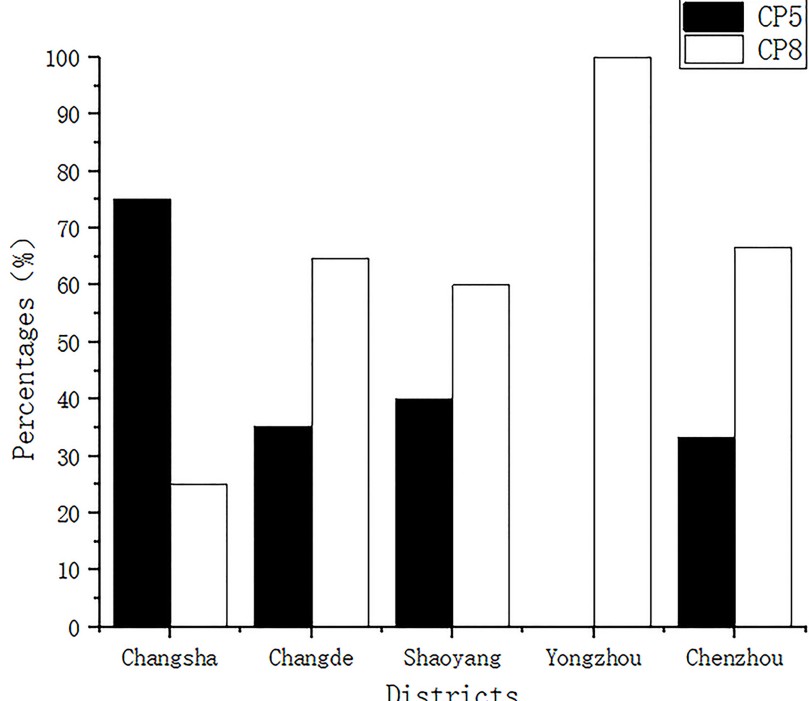

**Figure 1  Capsular polysaccharide detection.** All isolates were classified into capsular polysaccharide serotypes 5 and 8 (CP5 and CP8).

## Phenotypic characteristics of the isolates

The resistance of the 76 *S. aureus* strains to 18 antibiotics was tested using the broth dilution method. The resistance breakpoints of the 18 antibiotics and positive antibiotic

**Table 3  Antibiotics resistance of _Staphylococcus aureus_ isolates.** The breakpoint of antimicrobial resistance was judged according to the standard of the Clinical and Laboratory Standards Institute. The isolates resistant to oxacillin and vancomycin were verified by three repeated tests. OS-MRSA, oxacillin-sensitive mecA-positive _S. aureus_; OR-MRSA, oxacillin-resistant mecA-positive _S. aureus_.

| | Antibiotics | Resistance MIC ($\mu$g mL$^{-1}$) | No (%) of resistant isolates ($n = 76$) | | | Total |
|---|---|---|---|---|---|---|
| | | | OS-MRSA | OR-MRSA | MSSA | |
| $\beta$-lactams | Penicillin | $\geq 0.25$ | 19(25.00%) | 10(13.16%) | 34(44.74%) | 63(82.89%) |
| | Cefoxitin | $\geq 8$ | 2(2.63%) | 9(11.84%) | 6(7.89%) | 17(22.37%) |
| | Oxacillin | $\geq 4$ | 0(0.00%) | 10(13.16%) | 9(11.84%) | 19(25.00%) |
| | Ceftiofur | $\geq 8$ | 0(0.00%) | 0(0.00%) | 2(2.63%) | 2(2.63%) |
| | Augmentin | $\geq 8$ | 1(1.32%) | 1(1.32%) | 1(1.32%) | 3(3.95%) |
| Macrolides | Erythromycin | $\geq 8$ | 4(5.26%) | 5(6.58%) | 8(10.53%) | 17(22.37%) |
| | Tilmicosin | $\geq 32$ | 2(2.63%) | 8(10.53%) | 6(7.89%) | 16(21.05%) |
| Lincosamides | Clindamycin | $\geq 4$ | 2(2.63%) | 8(10.53%) | 7(9.21%) | 17(22.37%) |
| Quinolones | Ofloxacin | $\geq 4$ | 1(1.32%) | 0(0.00%) | 4(5.26%) | 5(6.58%) |
| | Enrofloxacin | $\geq 4$ | 1(1.32%) | 0(0.00%) | 2(2.63%) | 3(3.95%) |
| Tetracyclines | Doxycycline | $\geq 16$ | 0(0.00%) | 0(0.00%) | 0(0.00%) | 0(0.00%) |
| Aminoglycosides | Gentamicin | $\geq 32$ | 2(2.63%) | 8(10.52%) | 5(6.58%) | 15(19.73%) |
| Sulfonamides | Sulfisoxazole | $\geq 512$ | 0(0.00%) | 2(2.63%) | 1(1.32%) | 3(3.95%) |
| | Trimethoprim S-ulfamethoxazole | $\geq 4$ | 0(0.00%) | 4(5.26%) | 4(5.26%) | 8(10.53%) |
| Phenicols | Florfenicol | $\geq 8$ | 1(1.32%) | 0(0.00%) | 5(6.58%) | 6(7.89%) |
| Glycopeptides | Vancomycin | $\geq 16$ | 0(0.00%) | 1(1.32%) | 0(0.00%) | 1(1.32%) |
| Diterpenes | Tiamulin | $\geq 32$ | 0(0.00%) | 1(1.32%) | 1(1.32%) | 2(2.63%) |
| Oxazolidinones | Linezolid | $\geq 8$ | 0(0.00%) | 1(1.32%) | 1(1.32%) | 2(2.63%) |

resistance results are shown in Table 3. We found that 88.15% (67/76) of the isolates were resistant to at least one antibiotic, 32.89% (25/76) were resistant to three or more antibiotics (MDR strains), and the highest resistance was found among isolates that were resistant to six antibiotic classes. Nine isolates were not resistant to any antibiotic. All isolates were the most resistant to penicillin, with 82.89% showing penicillin resistance, followed by oxacillin (25.00%), erythromycin (22.37%), clindamycin (22.37%), tilmicosin (21.05%), gentamicin (19.73%), trimethoprim/sulfamethoxazole (10.53%), florfenicol (7.89%), ofloxacin (6.58%), amoxicillin/clavulanate (3.95%), enrofloxacin(3.95%), sulfisoxazole (3.95%), ceftiofur (2.63%), tiamulin(2.63%), and linezolid (2.63%). One isolate (1.32%) was resistant to vancomycin and was identified as VRSA. All isolates were sensitive to doxycycline. In this study, a total of 21 isolates were resistant to three or more classes of antibiotics, and the multiple resistance rate was 27.63%. The isolates were resistant to six kinds of antibiotics at most. No isolates sensitive to class 1 or 2 drugs or resistant to all antibiotics were detected. Therefore, the 21 multi-resistant strains could only be classified as MDR, and no isolates could be classified as XDR or PDR. The drug resistance profiles of all MDR isolates are shown in Table 4.

## Antibiotic resistance genes of the isolates

All 76 strains carried two or more antibiotic resistance genes, with a maximum of eight genes. All isolates carried _femB_ with a detection rate of 100%, followed by _ermB_ (80.26%),

**Table 4  Spectrum of multi-drug resistance.**

| Type of antimicrobial resistance | Number of strains(%) | Strain number | Antibiotic resistance spectrum |
|---|---|---|---|
| macrolides-$\beta$-lactam-quinolones-glycopeptides-diterpenoids-oxazolidinones | 1(4.76%) | SY-14 | PEN-ERY-OXA-VAN-TIA-TIL-LZD |
| macrolides-$\beta$-lactam-sulfonamides-aminoglycosides-lincosamides | 8(38.09%) | CD-15 | PEN-ERY-OXA-CLI-SF-SXT-TIL-GEN |
|  |  | CD-16 | PEN-OXA-CLI-CEF-CFX-OXA-SXT-TIL-GEN |
|  |  | CD-21 | PEN-OXA-CLI-CEF-CFX-OXA-SXT-TIL-GEN |
|  |  | CD-22 | PEN–CLI -CFX-SF-OXA-SXT-TIL-GEN |
|  |  | CD-23 | PEN–CLI -CFX-SF-OXA-SXT-TIL-GEN |
|  |  | CD-24 | PEN-OXA-CLI-CFX-SF-OXA-SXT-TIL-GEN |
|  |  | CD-25 | PEN-CFX-CLI-OXA-SXT-TIL-GEN |
|  |  | CD-26 | PEN–CLI -CFX-SF-OXA-SXT-TIL-GEN |
| macrolides-$\beta$-lactam-aminoglycosides-lincosamides | 5(23.80%) | CD-1 | PEN-ERY-CLI-CFX-TIL-GEN |
|  |  | CD-2 | PEN-ERY-CFX-CLI-OXA-TIL-GEN |
|  |  | CD-5 | PEN-ERY-CFX-CLI-OXA-TIL-GEN |
|  |  | CD-6 | CFX-CLI-ERY-A/C-PEN-OXA-TIL-GEN |
|  |  | CD-7 | PEN-ERY-CFX-CLI-A/C-TIL-GEN |
| macrolides-$\beta$-lactam-lincosamides-aminoglycosides | 2(9.52%) | CD-34 | PEN-ERY-CLI-CFX-OXA-GEN |
|  |  | SY-8 | PEN-CFX-CLI-ERY-OXA-TIL-GEN |
| quinolones-macrolides-$\beta$-lactam-sulfonamides | 1(4.76%) | CS-3 | PEN-ERY-OFL-ENR-SF |
| quinolones-macrolides-$\beta$-lactam | 2(9.52%) | CS-1 | PEN-ERY-OFL-EN |
|  |  | CS-2 | PEN-ERY-OFL-EN |
| macrolides-$\beta$-lactam-lincosamides | 1(4.76%) | CD-8 | PEN-ERY-CLI |
| chloramphenicol-diterpenoids-oxazolidinones | 1(4.76%) | YZ-2 | FFC-TIA-LZD |

*aacA-aphD* (76.32%), *femA* (75.00%), *ermC* (68.42%), *mecA* (47.37%), *flor* (42.11%), and *tetM* (38.16%); however, no isolates carried the *glrA*, *ermA*, or *vanA* genes. In this study, only the detection of *mecA*, *ermC*, and *aacA-aphD* in the corresponding antimicrobial resistance phenotypes was statistically significant (Table 5). The findings related to 11 resistance genes for seven types of antibiotics are shown in Fig. 2.

## MLST

MLST was successfully performed for all ($n = 76$) isolates, and the results are summarized in Table 6 and Figs. 3 and 4. Twenty-two STs were detected, of which ST7 accounted for 23.68% (18/76), ST97 for 14.47%(11/76), ST188 for 9.21% (7/76), and ST398 for 6.57% (5/76) of the isolates. The remaining STs were diverse; six novel alleles and eight new STs were detected in this study, with ST7737, ST7738, ST7739, ST7740, ST7756, and ST7757 carrying new alleles (Fig. 5). Although no new alleles were found for ST7735 and ST7736, new arrangements of alleles existed and were also considered to be new STs. All novel alleles and STs were uploaded to the PubMLST database (https://pubMLST.org/). The

**Table 5    Comparison of genotypic and phenotypic characteristics for antimicrobial resistance in *Staphylococcus aureus* isolates.** The pheno-types of *β*-lactam resistance were based on penicillin, oxacillin, cefoxitin, ceftiofur, and augmentin, while those of macrolide-lincomide resistance were based on erythromycin, tilmicosin, and clindamycin; tetracycline resistance phenotype was based on doxycycline; aminoglycoside resistance phenotype was based on gentamicin; and chloramphenicol resistance phenotype was based on florfenicol. The association between resistance phe-notype and resistance genes was significant ($p < 0.05$).

| Antibiotics | Genes | Characteristics of *S. aureus* isolates | | Association $p$ |
|---|---|---|---|---|
| | | G+ No(%) | P+ No(%) | |
| *β*-lactams | *mecA* | 36(47.37%) | 63(82.89%) | 0.0002 |
| | *femA* | 57(75.00%) | | 0.861 |
| Macrolides-Lincoamines | *ermB* | 61(80.26%) | 25(32.89%) | 0.229 |
| | *ermC* | 52(68.42%) | | 0.011 |
| Tetracyclines | *tetM* | 29(38.16%) | 0(0.00%) | 1.000 |
| Aminoglycosides | *aacA-aphD* | 58(76.32%) | 15(19.73%) | 0.017 |
| Phenicols | *Flor* | 32(42.11%) | 6(7.89%) | 0.685 |

**Notes.**
G+ resistance gene positive
P+ phenotypic resistance

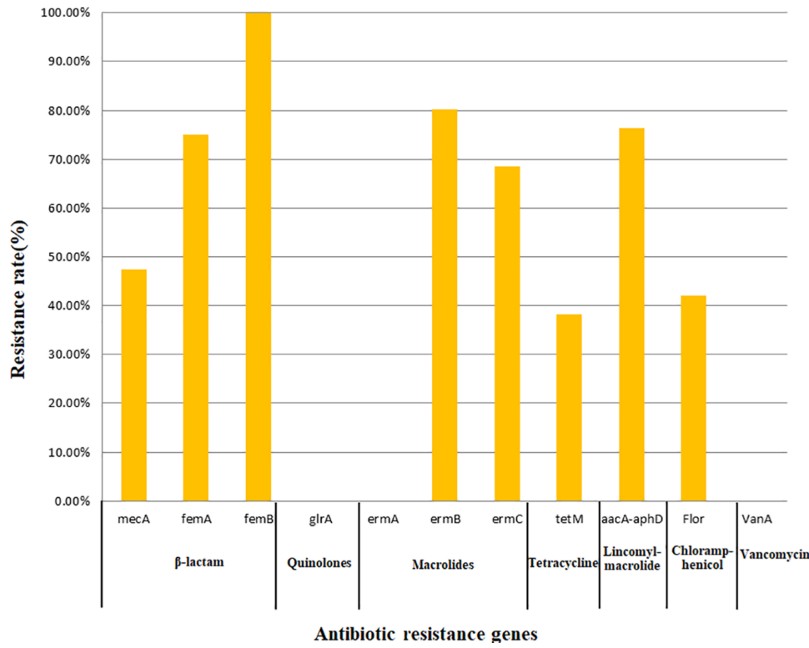

**Figure 2    Detection rate of antimicrobial resistance genes.**

Simpson's diversity index of the 76 isolates based on MLST was 0.907, which conformed to a 95% confidence interval (0.869–0.946). Clustering these STs using the eBURST algorithm revealed that ST97, ST7735, and ST7756 belonged to Clonal Complex (CC)97; ST1, ST81, ST188, and ST7737 belonged to CC1; ST398 belonged to CC398; ST5 and ST6 belonged to CC5; and ST2454 belonged to CC8.

**Table 6  MLST distribution of *Staphylococcus aureus* isolates.**

| STs[a] | clonal complex | No (%) of isolates | Allelic profiles[b] | | | | | | | Simpson's diversity index (95% CI[c]) |
|---|---|---|---|---|---|---|---|---|---|---|
| | | | arcC | aroE | glpF | gmk | pta | tpi | yqiL | |
| ST7 | | 18(23.68%) | 5 | 4 | 1 | 4 | 4 | 6 | 3 | |
| ST97 | 97 | 11(14.47%) | 3 | 1 | 1 | 1 | 1 | 5 | 3 | |
| ST188 | 1 | 7(9.21%) | 3 | 1 | 1 | 8 | 1 | 1 | 1 | |
| ST398 | 398 | 5(6.57%) | 3 | 35 | 19 | 2 | 20 | 26 | 39 | |
| ST705 | | 4(5.26%) | 6 | 72 | 50 | 43 | 49 | 67 | 59 | |
| ST7735[a] | 97 | 3(3.94%) | 3 | 1 | 605 | 1 | 1 | 5 | 3 | |
| ST7738[a] | | 3(3.94%) | 5 | 1084[b] | 1 | 4 | 4 | 6 | 3 | |
| ST7756[a] | 97 | 3(3.94%) | 3 | 1 | 1 | 1 | 1 | 843[b] | 3 | |
| ST7757[a] | | 3(3.94%) | 6 | 125 | 213 | 27 | 13 | 13 | 1015[b] | |
| ST1 | 1 | 3(3.94%) | 1 | 1 | 1 | 1 | 1 | 1 | 1 | |
| ST81 | 1 | 2(2.63%) | 1 | 1 | 1 | 9 | 1 | 1 | 1 | 0.907(0.869-0.946) |
| ST479 | | 2(2.63%) | 52 | 79 | 54 | 18 | 56 | 32 | 65 | |
| ST1921 | | 2(2.63%) | 4 | 9 | 1 | 8 | 1 | 10 | 218 | |
| ST7740[a] | | 2(2.63%) | 5 | 4 | 1 | 599[b] | 4 | 6 | 3 | |
| ST5 | 5 | 1(1.31%) | 1 | 4 | 1 | 4 | 12 | 1 | 10 | |
| ST6 | 5 | 1(1.31%) | 12 | 4 | 1 | 4 | 12 | 1 | 3 | |
| ST629 | | 1(1.31%) | 12 | 89 | 1 | 1 | 4 | 5 | 90 | |
| ST1458 | | 1(1.31%) | 5 | 4 | 1 | 4 | 4 | 6 | 39 | |
| ST2454 | 8 | 1(1.31%) | 3 | 3 | 1 | 1 | 264 | 1 | 10 | |
| ST7736[a] | | 1(1.31%) | 3 | 35 | 19 | 1 | 1 | 26 | 39 | |
| ST7737[a] | 1 | 1(1.31%) | 899[b] | 1 | 1 | 9 | 1 | 1 | 1 | |
| ST7739[a] | | 1(1.31%) | 18 | 95 | 941[b] | 2 | 7 | 15 | 5 | |

**Notes.**
[a] Novel ST types.
[b] Novel alleles.
CI, confidence interval; MLST, multi-locus sequence typing; ST, sequence typing.

### *spa* typing

Twenty *spa* types were found among the 76 strains (Table 7). The most common type was t796 (23.68%, 18/76), followed by t189(10.52%, 8/76), t359 and t091 (9.21%, 7/76 each), t034 (6.57%, 5/76), and t529 (5.26%, 4/76). The remaining *spa* types were found in three or fewer strains and showed a scattered distribution. The Simpson's diversity index of the *spa* types of the 76 isolates was 0.910, and the value conformed to a 95% confidence interval (0.873–0.946).

### Correlation between *spa*-MLST-mapping and antimicrobial resistance

ST7 was found in t796, t2383, and t091; ST97 was found in t267, t359, and t189; and ST705 was found in t701 and t529. Three STs were present in t796 and two in t529, whereas the remaining STs corresponded to only one *spa* type. Statistical analyses revealed a significant relevance between *spa* typing and MLST ($p = 0.001$, $< 0.05$).

Oxacillin-resistant MRSA (OR-MRSA) strains mainly expressed t796-ST7 ($n = 7$), and 93.33% t796-ST7 strains were MDR. The antibiotic resistance rates of cefoxitin, clindamycin, gentamicin, and temicoxin were all >70%, and those of cefoxitin, temicoxin,

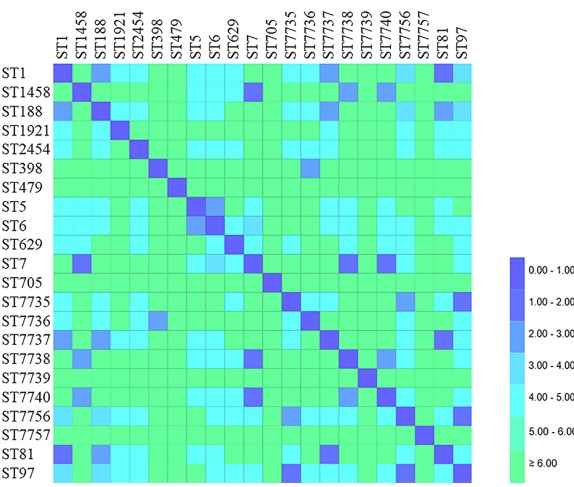

**Figure 3** **Distance matrix showing genetic relationship between classes of sequence typing (ST).** Each horizontal row and column represent a class of sequence typing (ST). The color of each coordinate indicates the distance of the genetic relationship between two STs. The closer the genetic relationship is the smaller the allele difference is, and the smaller the corresponding distance difference is. The ST difference in the same clonal complex group is less than 2.00.

**Table 7** *Spa* types distribution of *Staphylococcus aureus* isolates.

| *Spa* types | No. of isolates | Repeat succession | Simpson's diversity index (95% CI) |
|---|---|---|---|
| t796 | 18(23.68%) | 07-23-21-17-12-23-02-12-23 | |
| t189 | 8(10.52%) | 07-23-12-21-17-34 | |
| t359 | 7(9.21%) | 07-23-12-21-17-34-34-33-34 | |
| t091 | 7(9.21%0 | 07-23-21-17-34-12-23-02-12-23 | |
| t034 | 5(6.57%) | 08-16-02-25-02-25-34-24-25 | |
| t529 | 4(5.26%) | 04-34 | |
| t267 | 3(3.94%) | 07-23-12-21-17-34-34-34-33-34 | |
| t2383 | 3(3.94%) | 08-16 | |
| t18636 | 3(3.94%) | 121-21-16-34-17-82-24-17-17 | |
| t114 | 3(3.94%) | 07-16-34-33-13 | |
| t934 | 3(3.94%) | 07-23-12-34-34-34-34-33-34 | 0.910 (0.873-0.946) |
| t543 | 2(2.63%) | 04-20-17 | |
| t5100 | 2(2.63%) | 07-21-16-34-33-13 | |
| t164 | 2(2.63%) | 07-06-17-21-34-34-22-34 | |
| t7295 | 1(1.31%) | 04-31-17-17 | |
| t701 | 1(1.31%) | 11-10-21-17-34-24-34-22-25-25 | |
| t2700 | 1(1.31%) | 07-16-12-23-02-12-23-02-02-34 | |
| t2247 | 1(1.31%) | 26-25-17 | |
| t19817 | 1(1.31%) | 07-21-16-34 | |
| t002 | 1(1.31%) | 26-23-17-34-17-20-17-12-17-16 | |
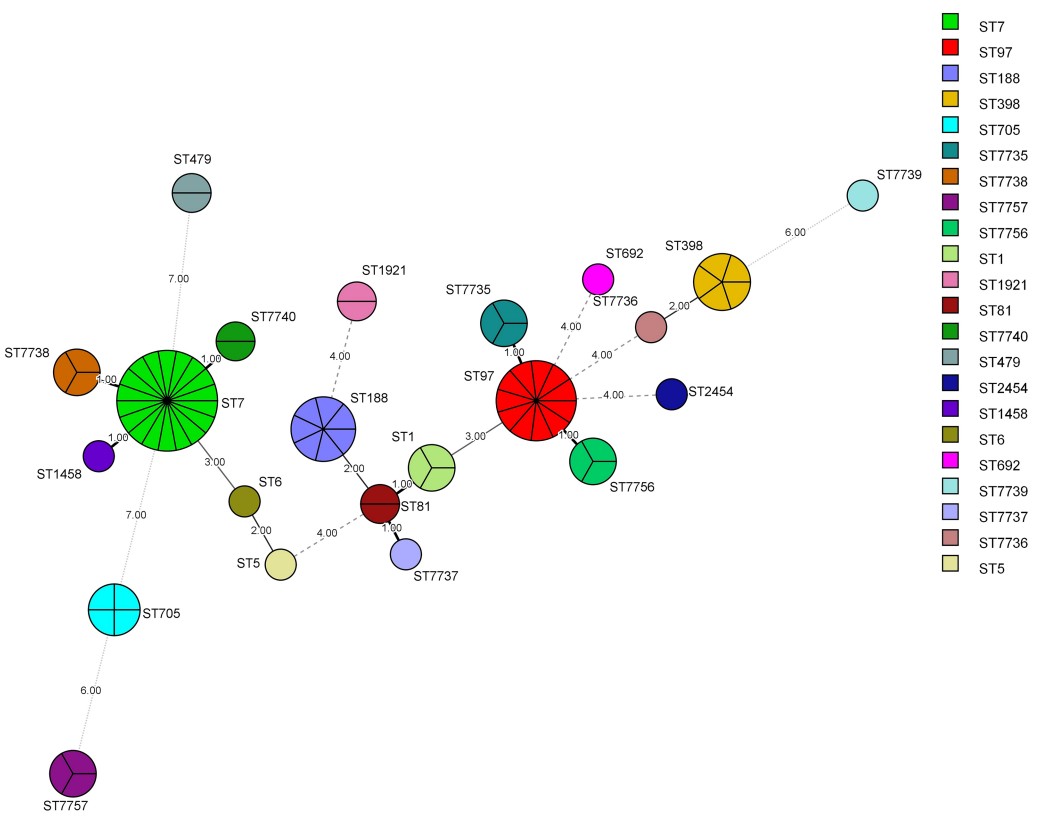

**Figure 4  Minimum Spanning Tree (MST).** The nodes of different colors represent different MLST types, and the size represents the relative number of isolates. The values between the lines represent the number of different alleles between two adjacent STs, and nodes with allelic differences of more than 4 are connected by dotted lines. The STs from common CCs had at least five same alleles.

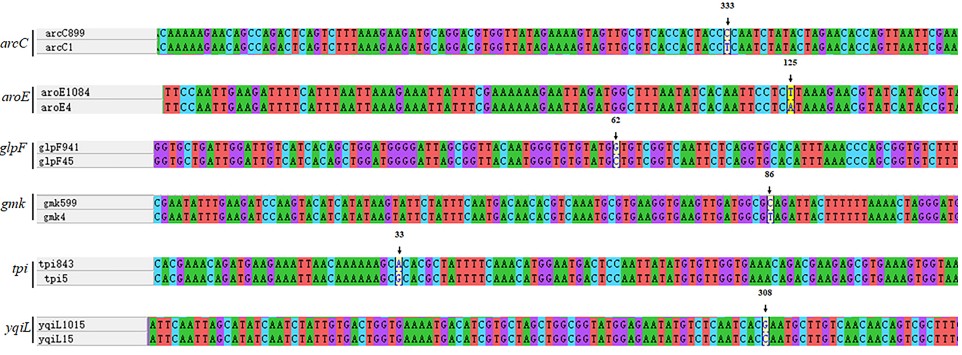

**Figure 5  Analysis of nucleic acid sequence of novel alleles.** Every new allele was compared to its closest matches. ST7737 *arcC*:333T →C; ST7738 *aroE*:125A →T; ST7739 *glpF*:62C →G; ST7740 *gmk*:86T →C; ST7756 *tpi*:33G →A; ST7757 *yqiL*:308C →G.

and cotrimoxazole were significantly higher than those of other molecular types ($p < 0.05$). The t934-CC97-ST7735 isolates were generally resistant to penicillin and erythromycin and 100% resistant to quinolones; the resistance rate to quinolones was significantly higher than that of other molecular types ($p < 0.05$). The isolates of t796-ST7740 were OR-MRSA and resistant to penicillin, benzoxicillin, and cefoxitin. In addition, other CC97 and CC398 isolates were generally resistant to penicillin, but no MDR strains were detected. The detection rate of OS-MRSA was relatively high in ST97 (45.45%), which may be the main molecular type of OS-MRSA from dairy cows in Hunan Province. The number of isolates with molecular type t189-CC1-ST188 was relatively large, but the isolates showed no resistance to all drugs except for two strains that were only resistant to penicillin. None of the t18636-ST7757 isolates showed resistance to any antimicrobial agent. The distribution of the molecular types and specific drug resistance phenotypes of the isolates is presented in Table 8.

# DISCUSSION

The isolation rate of *S. aureus* from raw milk in Hunan Province was 11.16%, which is lower than those observed from milk sources in northern Xinjiang, China (43.1%), by *Kou et al. (2021)* and bulk canned milk samples from Shandong Province, China (28.9%), by *Zhao et al. (2021)*. However, our results are similar to those of *Bissong et al. (2020)*, who reported a prevalence of 11% for *S. aureus* in beef and milk in Cameroon. *S. aureus*, an important multi-host pathogen, can be isolated from dairy cows and contaminated milk (*Gurjar et al., 2012*; *Fetsch et al., 2014*). It can spread between people and livestock during grazing, activities involving human–animal interactions, and movement of herds between farms (*Zadoks et al., 2011*). Although the isolation rate of *S. aureus* from raw milk in Hunan Province is not high, these bacteria were successfully isolated from the samples collected, suggesting that farmers should strictly control the cleaning and disinfection of the feeding environment of dairy cows. In addition, individuals involved in the dairy industry should implement standardized procedures for raw milk collection, processing, and packaging and take precautionary measures to reduce environmental pollution and avoid storing dairy products in unfavorable conditions to safeguard the health of consumers.

Serotyping of *S. aureus* CPs is usually performed to predict and determine the epidemiologic trends of strains isolated in an area and the incidence of mastitis in dairy cows before molecular typing (*Poutrel et al., 1988*). In this study, all *S. aureus* isolates exhibited either the CP5 or CP8 type; the overall detection rate of CP8 was higher than that of CP5, indicating that CP8 was the main epidemic serotype of *S. aureus* isolated in Hunan Province. *Poutrel et al. (1988)* found that CP5 was the main epidemic serotype in France. In a study by *Han, Pak & Guidry (2000)* in South Korea, the detection rate of CP5 was similar to that of CP8, and 26% of the strains were of the CP5 type and 24.2% of the CP8 type. The serotype distribution of CPs differs between countries and regions, indicating that the distribution of CPs of *S. aureus* is likely related to geographical location. In addition to serotyping *S. aureus*, CPs can help identify the antibody induced by its protein-coupled vaccine, which has immunogenicity and has been proven to protect mice from fatal *S.*

**Table 8  Analysis of *spa*-MLST-mapping and MRSA.**

| *Spa* typing | STs(CCs) | No. of resistant isolates(%) | | |
|---|---|---|---|---|
| | | OS-MRSA[b] | OR-MRSA[c] | MSSA |
| t934 | CC97-ST7735 | 1(1.31%) | 0(0.00%) | 2(2.63%) |
| | ST7 | 2(2.63%) | 7(9.21%) | 6(7.89%) |
| t796 | ST7756 | 1(1.31%) | 0(0.00%) | 0(0.00%) |
| | ST7740 | 0(0.00%) | 2(2.63%) | 0(0.00%) |
| t7295 | ST7739 | 0(0.00%) | 1(1.31%) | 0(0.00%) |
| t701 | ST705 | 0(0.00%) | 0(0.00%) | 1(1.31%) |
| t543 | ST479 | 1(1.31%) | 0(0.00%) | 1(1.31%) |
| t529 | ST705 | 1(1.31%) | 0(0.00%) | 2(2.63%) |
| | CC5-ST6 | 1(1.31%) | 0(0.00%) | 0(0.00%) |
| t5100 | CC1-ST81 | 0(0.00%) | 0(0.00%) | 2(2.63%) |
| t359 | CC97-ST97 | 2(2.63%) | 0(0.00%) | 5(6.57%) |
| t2700 | CC8-ST2454 | 0(0.00%) | 0(0.00%) | 1(1.31%) |
| t267 | CC97-ST97 | 2(2.63%) | 0(0.00%) | 1(1.31%) |
| | ST7 | 0(0.00%) | 0(0.00%) | 1(1.31%) |
| t2383 | ST1458 | 0(0.00%) | 0(0.00%) | 1(1.31%) |
| | ST7736 | 1(1.31%) | 0(0.00%) | 0(0.00%) |
| t2247 | ST692 | 1(1.31%) | 0(0.00%) | 0(0.00%) |
| t19817 | CC1-ST7737 | 0(0.00%) | 0(0.00%) | 1(1.31%) |
| t189 | CC1-ST188 | 3(3.94%) | 0(0.00%) | 4(5.26%) |
| | CC97-ST97 | 1(1.31%) | 0(0.00%) | 0(0.00%) |
| t18636 | ST7757 | 3(3.94%) | 0(0.00%) | 0(0.00%) |
| t164 | ST1921 | 1(1.31%) | 0(0.00%) | 1(1.31%) |
| t114 | CC1-ST1 | 3(3.94%) | 0(0.00%) | 0(0.00%) |
| | ST7 | 0(0.00%) | 0(0.00%) | 2(2.63%) |
| t091 | ST7738 | 0(0.00%) | 0(0.00%) | 3(3.94%) |
| | CC97-ST7756 | 0(0.00%) | 0(0.00%) | 2(2.63%) |
| t034 | CC398-ST398 | 2(2.63%) | 0(0.00%) | 3(3.94%) |
| t002 | ST5 | 0(0.00%) | 0(0.00%) | 1(1.31%) |

*aureus* infection (*Han, Pak & Guidry, 2000*). A targeted vaccine developed according to the prevalence of CP serotypes in particular areas can yield an improved immune effect.

Many antibiotics are used in the animal breeding industry as therapeutics or growth enhancers every year in China. However, with the evolution of bacteria and abuse of antibiotics, the antibiotic resistance rate of *S. aureus* has gradually increased. It is generally believed that the antibiotic resistance mechanism of *S. aureus* mainly includes the following: (1) drug hydrolase and modification (such as the *aacA-aphD* gene code modifies specific groups of aminoglycosides to make them lose their affinity for bacterial target ribosomes) (*Wang et al., 2015*); (2) the active efflux system pumps drugs out of bacterial cells (for example, the transporter encoded by *msr* gene can expel intracellular lincoamines to reduce intracellular drug concentration); and (3) changes in the target sites of antibacterial drugs (such as *erm* gene encoding methylase, which methylates the target of macrolide

drug action and reduces its affinity with bacterial ribosomes) (*Di Modugno et al., 2002*). In this study, *S. aureus* isolated from raw milk from Hunan Province was highly resistant to $\beta$-lactam antibiotics, with 82.89% of the isolates were resistant to penicillin. This finding is similar to that of studies from Shandong (74.4%) (*Zhao et al., 2021*) and the Ningxia Hui Autonomous Region (94.3%) (*Wang et al., 2016*), suggesting that *S. aureus* isolated from Chinese dairy cows is generally resistant to penicillin. The mechanism of penicillin resistance in *S. aureus* is mainly through the synthesis of $\beta$-lactam hydrolase, which hydrolyzes the $\beta$-lactam ring, or *mecA*-encoded PBP2a, which does not easily bind with $\beta$-lactams, to replace the effect of transpeptidase PBPs, thus inhibiting the ability of $\beta$-lactams to interfere with bacterial cell wall synthesis and reducing the antimicrobial effect of $\beta$-lactam drugs (*Hassanzadeh et al., 2020*). Of the 76 isolates, 36 were found to carry *mecA* and were identified as MRSA, of which 27.77% were OR-MRSA and 72.22% were OS-MRSA. As *Quijada et al. (2019)* showed in their study, deletions or mutations in the *bla* system (*blaI-blaR1-blaZ*) of some isolates may downregulate the expression of *mecA*, thereby affecting the resistance of the MRSA strain to oxacillin and resulting in OS-MRSA. In addition, it was also found that 22.50% (9/40) of MSSA isolates were resistant to oxacillin, and *mecA*- negative but oxacillin resistance-positive phenotypes have been reported in India (*Kumar, Yadav & Singh, 2010*) and Turkey (*Turutoglu et al., 2009*). This is possibly due to the role of PBP2c, which is expressed by *mecC* in some strains; overexpression of $\beta$-lactamase in borderline oxacillin-resistant *S. aureus*; or mutations at some amino acid sites in the PBP1, PBP2, and PBP3 transpeptidase domains, which enhance the resistance of bacteria to $\beta$-lactam antibiotics, resulting in oxacillin resistance despite *mec* gene negativity (*Hryniewicz & Garbacz, 2017*). *S. aureus* isolates were also resistant to macrolides, lincosamides, aminoglycosides, florfenicol, and quinolones, which may be related to the use of these drugs as the main antibiotics for treating bovine mastitis and other diseases at some dairy farms. Nevertheless, there is usually no significant correlation between antibiotic resistance genes and phenotypes (*Bissong & Ateba, 2020*) because bacterial antibiotic resistance phenotypes may be determined by a variety of factors, such as environmental changes, strain activity, and frequent use of antimicrobial agents, whereas genotypes are decided only by internal gene factors. In this study, a strain of MRSA was also found to be resistant to vancomycin. The strain was identified as VRSA, but *vanA* was not detected. *Bhattacharyya et al. (2016)* found that the emergence of VRSA was not completely related to *vanA* positivity, and *Roch et al. (2014)* proved that *S. aureus* could trigger the vancomycin-resistant phenotype through exposure to non-glycopeptide antibiotics, such as $\beta$-lactams. The development of vancomycin resistance in *S. aureus* could also be related to epigenetic processes and not only to gene expression (*Haaber et al., 2015*), which could explain why the only strain of VRSA in this study was also an MDR OR-MRSA.

A large international database containing the sequence data of thousands of *S. aureus* strains has been established. It is an important data source for large-scale epidemiological and genetic correlation research on *S. aureus* (*Van Wamel et al., 2010*). In this study, we found abundant and diverse STs of *S. aureus* isolated from raw milk in Hunan Province, with ST7 being the most common. Some studies have shown that ST398 is the main

ST related to livestock infections, and ST5, 45, and 239 are related to hospital-acquired infections. ST1, 8, 80, and 59 are mainly associated with community-acquired infection (*Skov & Jensen, 2009*), while ST97 is commonly associated with *S. aureus-* induced mastitis worldwide (*Smith et al., 2005*). In this study, CC398-ST398, CC5-ST5, CC1-ST1, and CC97-ST97 were all detected, which further confirmed the threat posed by raw milk infected with *S. aureus* to the dairy industry and public health. It is inferred that pathogenic bacteria may spread to the environment and infect humans during milking and transportation and *via* inadequately sterilized dairy products. Additionally, a previous study reported that alleles having only a single nucleotide mutation should be labeled as a novel ST (*Enright et al., 2000*). In this study, CC1-ST7737, ST7738, ST7739, ST7740, CC97-ST7756, and ST7757 showed altered nucleotide sequences, but only CC97-ST7735 and ST7736 showed a new allele arrangement. The emergence of these new STs is not accidental, and related mutants might have appeared a long time ago and spread among dairy cows in Hunan Province. However, these new STs were only discovered recently, possibly due to the lack of research on the molecular typing of *S. aureus* in raw milk from Hunan. The detection rate of MRSA in the 17 strains with newly discovered STs was 52.94%, of which ST7738 and 7740 were MRSA, indicating that these new STs may become an important molecular type of MRSA in Hunan Province. Furthermore, the only VRSA strain exhibited ST7739, indicating that the monitoring of new STs from isolates obtained during epidemics using MLST should be strengthened.

Twenty *S. aureus spa* types were identified in this study. Diversity analysis showed that the diversity of *spa* typing was high; t796 was the most common, followed by t189, t359, and t091. *Mekonnen et al. (2018)* identified 20 *spa* types from 79 strains of *S. aureus* isolated from milk samples from dairy farms in northwest Ethiopia, and t042 was the main *spa* type. *Strommenger et al. (2006)* studied *S. aureus* isolated from Dutch cows with mastitis and found that t529 was the dominant *spa* type. Upon comparing the above results, we speculate that isolates from different countries and regions usually have different molecular types. Many studies have used MLST and *spa* typing to determine the population structure of highly cloned *S. aureus* (*Bhargava et al., 2011*; *Basanisi et al., 2017*). *Malachowa et al. (2005)* used pulse field gel electrophoresis, MLST, multiple locus variable-number tandem repeat analysis, and *spa* typing to compare 59 strains of iatrogenic *S. aureus* and found that *spa* typing showed similar results to those of MLST. They suggested that multiple typing methods should be integrated to improve the accuracy of *S. aureus* typing. In this study, there was a significant correlation between *spa* typing and MLST. Although *spa* types and STs with multiple corresponding genotypes were observed, most STs and *spa* types corresponded to 1 genotype. The existence of different STs associated with the same *spa* type indicates that 2 STs may have originated from the same clone and evolved in parallel (*Strommenger et al., 2008*). The *spa*-MLST-mapping typing of the 76 isolates clearly reflected the molecular characteristics of *S. aureus* isolates in Hunan Province and that t796-ST7 was the main molecular type of *S. aureus*. Notably, five isolates exhibited t034-CC398-ST398, which has been reported to be the main molecular type of *S. aureus* infection in livestock and poultry (*Bouiller et al., 2016*), and MRSA isolates exhibiting t034-CC398-ST398 have been discovered in domestic pigs (*Graveland et al., 2011*). It is

speculated that t034-CC398-ST398 isolates may spread among different kinds of livestock in different regions, and they may even spread between livestock and humans and cause diseases (*Van Cleef et al., 2011*); therefore, farmers should be vigilant.

Through the isolation and identification of milk-derived *S. aureus* in northern Italy, *Gazzola et al. (2020)* found that CC398, CC1, and CC97 were three common MRSA clones, and *Li et al. (2015)* found that the main molecular type of MRSA isolated from milk in Shaanxi Province was t524-ST71. These studies suggest that the distribution of MRSA may be related to its molecular typing. In this study, different molecular isolates showed different antibiotic resistance characteristics, and the most common molecular type exhibited by MRSA was t796-ST7. The isolates of the same molecular type may have the same or similar antibiotic resistance spectrum. The isolates obtained from different regions or sources may have different molecular types, and the isolates of different molecular types may have different antibiotic resistance, which is speculated to result from the competition between different molecular types under the screening pressure of antibiotics in different regions, and the prevalence of dominant molecular types can promote further spread of antibiotic-resistant bacteria. In addition, *Enright et al. (2000)* pointed out that it is usually impossible for unrelated MRSA and MSSA to have the same molecular typing and that this status may be caused by the horizontal transfer of *mecA* from MRSA to MSSA or the loss or inactivation of the *mec* gene of some MRSA strains, resulting in some MRSA strains acquiring the same allele map as their MSSA ancestors and thus being classified as having the same STs.

## CONCLUSIONS

The results showed that *S. aureus* from raw milk in Hunan area had a typical CP type, rich molecular typing, and different levels of antibiotic resistance to a variety of antibiotics. Our findings could help promote the production of *S. aureus* vaccine against CP8 in Hunan area and resistance monitoring of t796-ST7 isolates with the prevalent molecular type of MDR strains. The present results suggest reducing the use of $\beta$-lactam, macrolide, lincoamine antibacterial drugs and appropriately administering doxycycline, sulfonamides, and glycopeptides as veterinary antibiotics to treat infectious diseases in dairy cows. The antibiotic drug dosage should be strictly controlled, and drug rotation and the withdrawal period of the antibiotics should be strictly implemented. Further, active research, development, and promotion of Chinese herbal medicines, micro-ecological preparations, and other new non-antibiotic drugs are necessary.

### Funding

This research was funded by the Science Research Project of the Hunan Education Department and the Excellent Youth Project, grant number 20B290. The funders had no role in study design, data collection and analysis, decision to publish, or preparation of the manuscript.

## Grant Disclosures

The following grant information was disclosed by the authors:
Science Research Project of the Hunan Education Department and the Excellent Youth Project: 20B290.

## Competing Interests

The authors declare there are no competing interests.

## Author Contributions

- Keming Ning conceived and designed the experiments, performed the experiments, analyzed the data, prepared figures and/or tables, authored or reviewed drafts of the article, and approved the final draft.
- Rushun Zhou performed the experiments, analyzed the data, prepared figures and/or tables, authored or reviewed drafts of the article, and approved the final draft.
- Manxiang Li conceived and designed the experiments, performed the experiments, analyzed the data, prepared figures and/or tables, authored or reviewed drafts of the article, and approved the final draft.

## DNA Deposition

The following information was supplied regarding the deposition of DNA sequences:

Data are available at PubMLST: arcC: 899, aroE: 1084, glpF: 941, gmk: 599, tpi: 843, yqiL: 1015

https://pubmlst.org/bigsdb?db=pubmlst_saureus_seqdef&page=alleleInfo&locus=arcC&allele_id=899,

https://pubmlst.org/bigsdb?db=pubmlst_saureus_seqdef&page=alleleInfo&locus=aroE&allele_id=1084

https://pubmlst.org/bigsdb?db=pubmlst_saureus_seqdef&page=alleleInfo&locus=glpF&allele_id=941

https://pubmlst.org/bigsdb?db=pubmlst_saureus_seqdef&page=alleleInfo&locus=gmk&allele_id=599

https://pubmlst.org/bigsdb?db=pubmlst_saureus_seqdef&page=alleleInfo&locus=tpi&allele_id=843

https://pubmlst.org/bigsdb?db=pubmlst_saureus_seqdef&page=alleleInfo&locus=yqiL&allele_id=1015

## Data Availability

Raw data are available in the Supplemental Files.

## Supplemental Information

Supplemental information for this article can be found online at http://dx.doi.org/10.7717/peerj.15847#supplemental-information.

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
