# Peer review of "Antimicrobial resistance and molecular typing of Staphylococcus aureus isolates from raw milk in Hunan Province"

_PeerJ, doi:10.7717/peerj.15847_

## Round 0.1 · original submission · Major Revisions

Dear Dr. Ning and colleagues:

Thanks for submitting your manuscript to PeerJ. I have now received two independent reviews of your work, and as you will see, the reviewers raised some concerns about the research. Despite this, these reviewers are optimistic about your work and the potential impact it will have on research studying Staphylococcus aureus dynamics in raw milk. Thus, I encourage you to revise your manuscript, accordingly, taking into account all of the concerns raised by both reviewers.

Please provide more information to support CP5 and CP8 capsular types. There appears to be many missing references that should be included within the framework of your study. Please enlist of a native English speaker to help with language and grammar. Also, take the advice of the reviewers for improving the clarity and organization of your manuscript.

There are other minor concerns raised by the reviewers. Please address all of these issues and ensure that all information regarding experimental design, gene and bacterial names, primers, etc. are provided and correct. Please strive to make your study repeatable. This especially includes providing all of the necessary information in the Materials and Methods.

Reviewer 2 has requested that you cite specific references. You may add them if you believe they are especially relevant. However, I do not expect you to include these citations, and if you do not include them, this will not influence my decision.


I look forward to seeing your revision, and thanks again for submitting your work to PeerJ.

Good luck with your revision,

-joe

Reviewer 1 ·

Basic reporting

The enclosed study by Li et al. is about the prevalence of capsular polysaccharide types CP5 and CP8 and antimicrobial resistance genes in Staphylococcus aureus isolates found in raw milk- Hunan, China.

With the increasing casualties due to AMR, it is a very interesting study. However, the manuscript cannot be accepted in the current form at it is in the preliminary stage.

Minor:
1) There some grammatical errors. The authors may have to look into this.
2) Since Figure 1 graph labels are in a different language, I am unable to interpret the data.
3) The reference given for the AMR cases are outdated in the following pages:
Line #112  S. aureus can …… The reference is from the year 2014
Line #117  The reference is from the year 2011

4) Line #139: Is it a repetition?

5) There is a detailed explain for MLST. However, no explanation for spa and how mlst and spa are together used for distinguishing the bacterial strains (for, t796-ST7). Detailed explanation is required.

Experimental design

1) My major concern is that there is no explanation on how the authors concluded the capsular type as CP5 or CP8. What are all the gene(s) that is specific for CP5 and CP8?

2) How did the authors eliminate the other serotypes? Since Figure 1 graph labels are in a different language, I am unable to interpret the data.

3) Line # 138: what is the definition for “suspected colony”?

4) Line # 152: After blasting the sequencing results, what are all the Accession IDs that came as a hit? There is no mention on this point.

5) Line #197: What is the reason to choose K independent samples method and a non-parametric test to find the correlation between spa type and mlst?

Validity of the findings

I am not very convinced about the way the authors justified that CP5 and CP8 are the capsular types that were found. Basically, I am not seeing a proper validation for that as the primer they have chosen is for the gene (there is no mention about which genes) specific for CP5 and CP8. Indeed, these may be common to some other serotype also. But, I am not seeing any explanation in this context.

Additional comments

None

Reviewer 2 ·

Basic reporting

No comment

Experimental design

Comments to authors:
-The current study is interesting; however, the authors should address the following comments to improve the quality of the manuscript:

-The manuscript should be revised for English editing and grammar mistakes.

- Please write the scientific names of bacterial pathogens and genes in the correct form all over the manuscript and the references section.


Title:
I think the work would benefit from the title that contains the main conclusion of the study (should be derived from the conclusion). Please modify the title.

Abstract:
- The abstract must illustrate the used methods and the most prevalent results (give more hints about methods and results). Besides, rephrase the aim of the work and the main conclusion of your findings.

-A graphical abstract is recommended (If possible).

- Add the full expression before the abbreviations.

-Introduction: (it needs to be more informative):

-Give a hint about the virulence factors and the mechanism of disease occurrence , and infecions caused by S. aureus.

- The authors should illustrate the public health importance concerning the emergence of multidrug-resistant (MDR) bacterial pathogens that reflect the necessity of new potent and safe antimicrobial agents. Several studies proved the widespread MDR- bacterial pathogens;
Authors could add the following paragraph:
Multidrug resistance has been increased all over the world that is considered a public health threat. Several recent investigations reported the emergence of multidrug-resistant bacterial pathogens from different origins that increase the necessity of the proper use of antibiotics. Besides, the routine application of the antimicrobial susceptibility testing to detect the antibiotic of choice as well as the screening of the emerging MDR strains. You are advised to cite the following valuable studies:

1. PMID: 33061472
2.PMID: 32497922
3.PMID: 32397408
4. PMID: 36819057
5. PMID: 36365013
6. PMID: 33603418
7. PMID: 32961293

- Illustrate the mechanism of action different virulence factors of S. aureus, especially PVL toxin.

-Rephrase the aim of the work to be clear and better sound.

Material and methods:

- Support all methods with updated specific references.

• Add the company, city, and country of the used chemicals and reagents.

- S. aureus isolation and identification:
Discuss in detail the methods of isolation and identification of S. aureus. Besides, specific references should be added.
• Add the company, city, and country of the used bacterial media and reagents that were used in the biochemical identification of isolates. Also, enumerate all used biochemical reactions.

-Antibiotic susceptibility testing:

-Please, explain in detail
•Add the names of the antimicrobial classes and enumerate the tested antibiotics.

•The authors are advised to classify the tested isolates to MDR , XDR, and PDR as described by Magiorakos et al.
Magiorakos AP, Srinivasan A, Carey RB, Carmeli Y, Falagas ME, Giske CG, et al. Multidrug-resistant, extensively drug-resistant and pandrug-resistant bacteria: An international expert proposal for interim standard definitions for acquired resistance. Clin Microbiol Infect. 2012; 18:268–81. doi:10.1111/j.1469-0691.2011.03570.x.

- The detection of virulence and antimicrobial resistance genes in the recovered isolates should be performed. Afterwards, the correlation between phenotypic and genotypic multidrug resistance should be performed.

-Detection of virulence genes is recommended (especially, the pvl gene).

-Statistical analyses: Add more details about the used software.

-Results:Good presentation:

- Please add a starting paragraph to the results section to briefly introduce the topic, your goals and
hypothesis and a short summary of what you did in this work.

-Add this subtitle: Phenotypic characteristics of the recovered isolates:

• Illustrate in detail the phenotypic characteristics of the recovered S. aureus
isolates.

-Antimicrobial susceptibility testing:

• -Illustrate in a new table the occurrence of MDR (Multidrug resistance) among the recovered isolates as the following (illustrate the names of the antimicrobial classes and different antibiotics):
No. of strains % Type of resistance
R, MDR, and XDR Phenotypic multidrug resistance
(Antimicrobial classes and different antibiotics). The antibiotic-resistance genes

-The correlation (Correlation coefficient) between phenotypic and genotypic multidrug resistance should be performed.


-Increase the resolution of all figures (must be 600 dpi).


-Discussion:

The authors are advised to illustrate the real impact of their findings without repetition of results.

Please illustrate different mechanisms of antibacterial resistance in S. aureus.

-Conclusion
- Should be rephrased to be sounded. A real conclusion should focus on the question or claim you articulated in your study, which resolution has been the main objective of your paper?

Validity of the findings

No comment

---

## Round 0.2 · Minor Revisions

Dear Dr. Ning and colleagues:

Thanks for revising your manuscript. The reviewer is mostly satisfied with your revision (as am I). Great! However, there are references to consider. Please add these or address why you will not ASAP so we may move towards acceptance of your work.

Best,

-joe

Reviewer 2 ·

Basic reporting

-The authors ignored the following important comment:

The authors should illustrate the public health importance concerning the emergence of multidrug-resistant (MDR) bacterial pathogens that reflect the necessity of new potent and safe antimicrobial agents. Several studies proved the widespread MDR- bacterial pathogens;
Authors could add the following paragraph:
Multidrug resistance has been increased all over the world that is considered a public health threat. Several recent investigations reported the emergence of multidrug-resistant bacterial pathogens from different origins that increase the necessity of the proper use of antibiotics. Besides, the routine application of the antimicrobial susceptibility testing to detect the antibiotic of choice as well as the screening of the emerging MDR strains. You are advised to cite the following valuable studies:

1. PMID: 33061472
2.PMID: 32497922
3.PMID: 32397408
4. PMID: 36819057
5. PMID: 36365013
6. PMID: 33603418
7. PMID: 32961293

Experimental design

No further comments.

Validity of the findings

No further comments

---

## Round 0.3 · accepted · Accept

Dear Dr. Ning and colleagues:

Thanks for revising your manuscript based on the concerns raised by the reviewers. I now believe that your manuscript is suitable for publication. Congratulations! I look forward to seeing this work in print, and I anticipate it being an important resource for groups studying Staphylococcus aureus dynamics in raw milk. Thanks again for choosing PeerJ to publish such important work.

Best,

-joe